# On the origin of the controversial electrostatic field effect in superconductors

I. Golokolenov [1,2,3], A. Guthrie [1], S. Kafanov [1✉], Yu. A. Pashkin [1✉] & V. Tsepelin [1]

Superconducting quantum devices offer numerous applications, from electrical metrology and magnetic sensing to energy-efficient high-end computing and advanced quantum information processing. The key elements of quantum circuits are (single and double) Josephson junctions controllable either by electric current or magnetic field. The voltage control, commonly used in semiconductor-based devices via the electrostatic field effect, would be far more versatile and practical. Hence, the field effect recently reported in superconducting devices may revolutionise the whole field of superconductor electronics provided it is confirmed. Here we show that the suppression of the critical current attributed to the field effect, can be explained by quasiparticle excitations in the constriction of superconducting devices. Our results demonstrate that a miniscule leakage current between the gate and the constriction of devices perfectly follows the Fowler-Nordheim model of electron field emission from a metal electrode and injects quasiparticles with energies sufficient to weaken or even suppress superconductivity.

[1] Department of Physics, Lancaster University, Lancaster, UK. [2] P. L. Kapitza Institute for Physical Problems of RAS, Moscow, Russia. [3] National Research University Higher School of Economics, Moscow, Russia. ✉email: sergey.kafanov@gmail.com; y.pashkin@lancaster.ac.uk

In semiconductor electronics, the field effect refers to the control of electrical conductivity in nanoscale devices[1], which underpins the field effect transistor, one of the cornerstones of present-day semiconductor technology[2]. The effect is enabled by the penetration of the electric field far into a weakly doped semiconductor, whose charge density is not sufficient to screen the field. On the contrary, the charge density in metals and superconductors is so large that the field decays exponentially from the surface and can penetrate only a short distance into the material. Hence, the field effect should not exist in such materials[3]. Nonetheless, recent publications have reported observation of the field effect in superconductors and proximised normal metal nanodevices[4–7]. The effect was discovered in gated nanoscale constrictions and nanowires as a suppression of the critical current under the application of intense electric field and interpreted in terms of an electric field-induced perturbation propagating inside the superconducting film. This interpretation is in stark contrast to the existence of the commonly accepted screening effect in metals.

Controlling the properties of superconducting films using gates was first proposed in the 1960s when it was found that electrostatic charging can affect the superconducting transition temperature of thin tin and indium films[8]. Several years later, a mesoscopic superconductor–normal metal–superconductor Josephson junction was predicted[9] and realised[10] by controlling the supercurrent flow via a "normal" current traversing the normal metal between the superconducting electrodes[11]. This control was attributed to the modified quasiparticle distribution, which was driven far from equilibrium by a voltage applied across the normal metal.

Another technique for controlling the Josephson supercurrent is to introduce a semiconductor in which carrier concentration can be tuned by electrostatic effects. As a result, a Josephson field-effect transistor (FET) was realised by building small hybrid superconductor–semiconductor structures where a region of sub-micron-long high-mobility two-dimensional electron gas (2DEG) was in good galvanic contact with two superconducting electrodes[12]. Through the development of nanofabrication techniques, it became possible to build such structures using various semiconductors and superconductors. A supercurrent through the whole structure was observed and controlled electrostatically by a nearby gate, due to the proximised superconductivity in the semiconductor[13–15]. Although at low voltages these devices act as Josephson junctions with a gate-controlled critical current, at high voltages they behave as conventional FETs.

In the later experiments, the 2DEG was replaced by indium arsenide semiconductor nanowires, with aluminium-based superconducting electrodes[16]. Below 1 K, owing to the proximity effect, the nanowires form superconducting weak links operating as mesoscopic Josephson junctions with electrically tunable coupling. A gate voltage controls the electron density in the nanowire, and regulates the supercurrent. Finally, the availability of semiconductor graphene flakes resulted in hybrid graphene/superconductor devices where the gate voltage controls supercurrent via either quasiparticles in the conduction band or quasiholes in the valence band[17].

Although the field effect in hybrid semiconductor–superconductor structures was predicted[12] and confirmed experimentally[13–15], it is not expected to exist in all-superconducting devices where the high carrier density screens the applied electric field. Therefore, the observation of the electrostatic field effect in metallic nanostructures[4–7] warrants further studies and interpretation.

In this work, we investigate a superconducting quarter-wave coplanar waveguide resonator shorted to the ground electrode through a gated nanoconstriction. Our results show that the application of high-gate voltages indeed reduces the resonance frequency of the resonator, corresponding to the suppression of the critical current, but, at the same time, it also results in the lower quality factor and higher noise intensity. We also measure a small leakage current between the gate and the constriction and find a strong correlation in the gate voltage dependence of this current and all other measured quantities. We conclude from our experiments that the observed reduction of the critical current can be explained by high-energy quasiparticle injection, which was also proposed in the recent experimental works[18,19].

## Results

**Device design and measurement setup.** Here, we investigated gated superconducting constrictions (see inset in Fig. 1a), identical to the structures described in refs. [4–7], in order to understand the origin of the observed field effect. The structure in the inset of

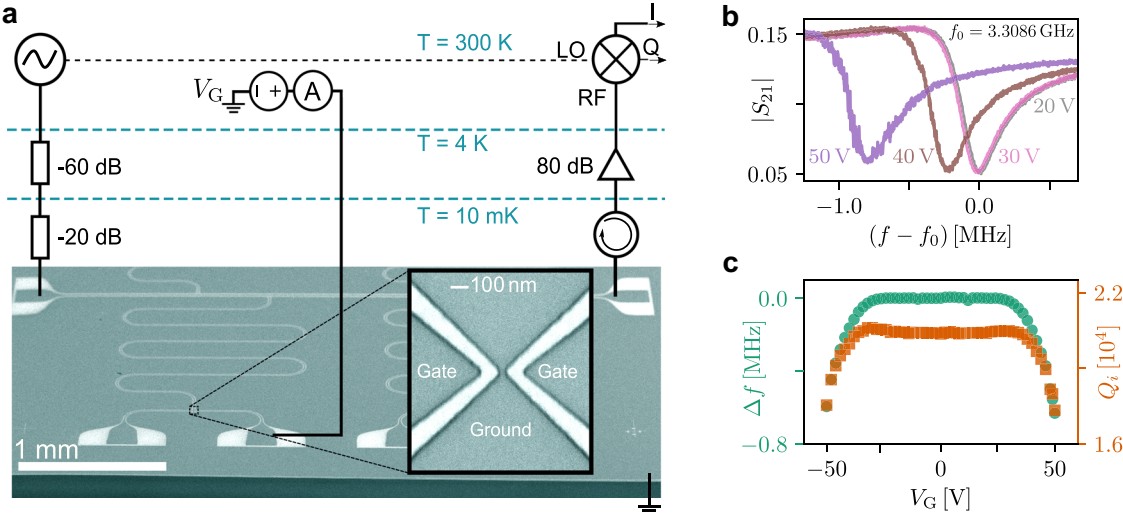

**Fig. 1 Experimental details and basic sample characterisation. a** Schematic of the measurement setup with an electron micrograph of the investigated superconducting structure consisting of a coplanar transmission line with four capacitively coupled quarter-wavelength microwave resonators. Each resonator is terminated to the ground plane by the constriction (Dayem bridge) shown in the inset. **b** A collection of the resonance curves at different gate voltages, $V_G$, for an investigated resonator. Note that the resonance frequency rapidly decreases at large gate voltages, $|V_G| > 30$ V. **c** The shift of the resonance frequency (left axis) and the internal quality factor of the resonator (right axis) as a function of applied gate voltage.

Fig. 1a is a Dayem bridge accompanied by two control gates located 100 nm apart on both sides of the constriction. The devices were formed in a 30 nm thick superconducting vanadium film ($T_c = 4.18\,K$) on the surface of oxidised undoped silicon. As the constriction has a characteristic dimension comparable with the vanadium superconducting coherence length $\xi_0 \approx 40\,nm$[20], it is a Josephson junction with inductance $L_J = \hbar/(2eI_c)$, where $I_c$ is the critical current of the constriction.

Figure 1a shows the schematic of our experiment and electron micrograph of the sample. To investigate the effect of the gate voltage on the superconductor, we have embedded each Dayem bridge into the current antinode of the quarter-wavelength coplanar microwave resonator, formed in the same vanadium film. A series of four meander-shaped resonators are incorporated into a manifold frequency multiplexing network (FMN)[21], which allows independent probing of each resonator at its resonance frequency using a single feedline. The coupling strength of each resonator to the feedline is weak, such that the quality factor is determined only by the resonators' internal losses.

The experiments were performed in a cryogen-free dilution refrigerator with a base temperature of 10 mK. The incoming microwave tone was filtered and attenuated at each temperature stage of the cryostat. After passing through the FMN, the transmitted signal was amplified by a series of cryogenic amplifiers and detected using an IQ demodulator, which allowed independent measurements of the in-phase, $I$, and quadrature, $Q$. The voltage on the control gates was applied through DC lines filtered at 10 mK. We used a biasing scheme to measure the leakage current, $I_L$, and differential conductance of the gap between the gate and the constriction, $dI_L/dV_G$.

**Response curves and quality factor**. All four resonators embedded into the FMN behaved alike, so here we present experimental results for one device with a resonance frequency $f_0 = 3.3086\,GHz$ measured at zero gate voltage. The resonance frequency, $\omega = (LC)^{-1/2}$, of the resonator is determined by its capacitance, $C$, and inductance, $L$. The internal $Q$-factor is given by $Q_i = \omega L/R$, where $R$ represents dissipation in the coplanar waveguide forming the resonator[22]. The total inductance is the sum of the resonator's geometric inductance, $L_g$, and the Josephson inductance, $L_J$, of the Dayem bridge. In our resonators, the geometric inductance is much larger than the Josephson inductance, i.e., $L_g \gg L_J$.

Figure 1b shows a collection of the resonance curves measured at various gate voltages, $|V_G| < 50\,V$, i.e., the same voltage range as has been used in the reported experiments[4–7]. The resonance frequency exhibits little change up to $|V_G| \approx 25\,V$, however, at higher $V_G$ it significantly decreases. The effect can be explained by the critical current suppression in the Dayem bridge, which leads to an increment of $L_J$ and reduction of the resonance frequency, since:

$$\omega \approx \frac{1}{\sqrt{L_g C}}\left(1 - \frac{1}{2}\frac{L_J}{L_g}\right). \tag{1}$$

The shift of the resonance frequency is bipolar and symmetric around zero gate voltage, see Fig. 1c (left axis). Our observations confirm the suppression of supercurrent in the Dayem bridge by the gate voltage[4–7] but disagree with the existence of the electrostatic field effect for the reasons presented below.

Figure 1c (right axis) demonstrates that the application of gate voltages within $|V_G| \approx 25\,V$ produces an expected small rise of the $Q$-factor caused by an increment of the Josephson inductance

$$Q_i \approx \frac{1}{R}\sqrt{\frac{L_g}{C}}\left(1 + \frac{1}{2}\frac{L_J}{L_g}\right). \tag{2}$$

However, $Q_i$ drops rapidly at higher voltages, which can only be caused by the increase of the internal losses of the resonator, i.e., dissipation, $R$. Note, that the bipolar and symmetric gate voltage dependence of the $Q$-factor is unexpected by itself for the negatively charged superconducting carriers, i.e., Cooper pairs. Furthermore, the resonance curves become visibly noisier at high-gate voltages, which also cannot be attributed to the stationary change of a reactive parameter, such as the Josephson inductance. Hence, all observations point towards higher dissipation in the resonator at high applied gate voltages and cast doubt on the detection of a straightforward field effect in superconducting constrictions.

**Noise properties**. We investigated the noise properties of the resonator in the same range of the applied gate voltages to understand the nature of the dissipation. The principle of the noise measurements is presented in Fig. 2a. The top and bottom panels show examples of the resonance curves for the magnitude of the transmitted microwave signal, for two gate voltages of 20 V and 50 V, correspondingly. After selecting the frequency of the working point (w.p.), corresponding to the steepest point of the

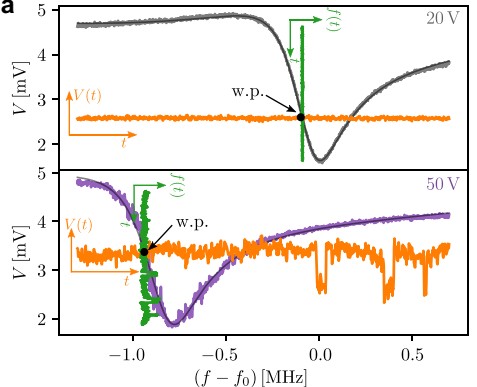
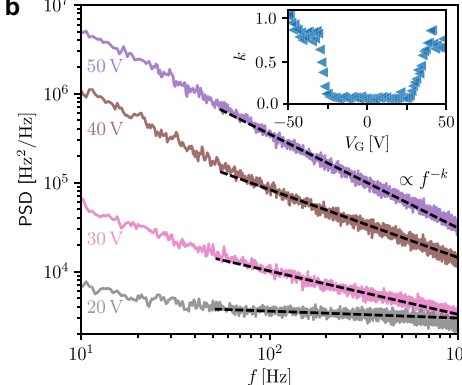

**Fig. 2 Noise properties of the superconducting coplanar resonator. a** Principle of the frequency noise measurements. Examples of resonance curves at 20 V (top panel) and 50 V (bottom panel) gate voltage applied to the Dayem bridge. At each gate voltage, the steepest point of the magnitude of the resonance is used as the working point (w.p.) for time trace measurements. Examples of voltage and frequency fluctuations with a duration of 1 s are shown in orange and green colours, respectively. **b** Power spectral density (PSD) of fluctuations at various gate voltages. Dashed lines are the fits of PSD functions by $\propto f^{-k}$. Inset shows the dependence of the fitted exponent $k$ on the gate voltage at 10 mK.

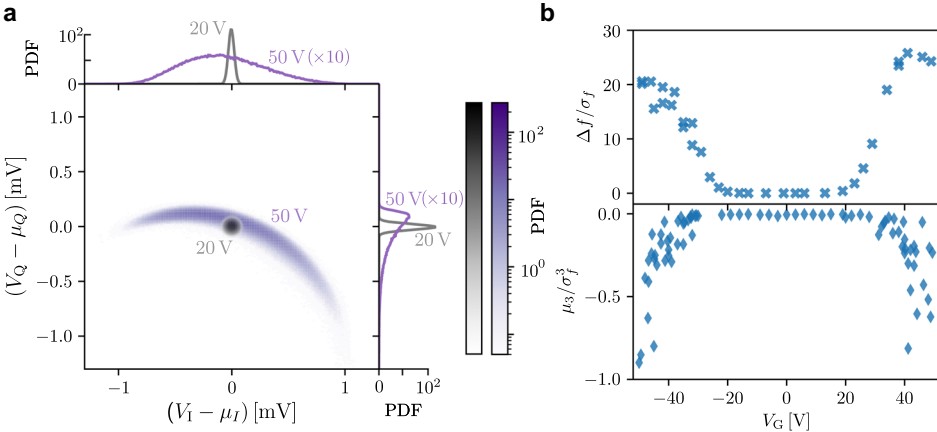

**Fig. 3 Histogram analysis of the voltage noise and reconstructed noise of the resonance frequency at 10 mK. a** An example of two-dimensional probability density function (PDF) of a normal deviate of the voltage noise obtained from the homodyne detection of the transmitted signal at a frequency corresponding to the w.p. in Fig. 2a for $V_G = 20$ V and $V_G = 50$ V. Side insets show the corresponding one-dimensional PDFs for $V_I$ and $V_Q$ components of the voltage noise for both values of the gate voltage. **b** Gate voltage dependence of the normalised frequency shift and standardised third-moment (skewness) of the frequency fluctuations. Both are normalised to the variance of the frequency fluctuations.

resonance curve, we drove the resonator at this frequency and detected the transmitted signal as a function of time. Figure 2a depicts an example of the measured time traces of the magnitude of the transmitted signal $V(t)$ in orange with a clear growth of fluctuations with larger $V_G$. The signatures of the telegraph-like noise visible at 50 V time trace start to appear at $|V_G| \approx 35$ V. The appearance and growth of the low-frequency telegraph noise with $V_G$ contradict the claim[4–7] of static suppression of super-conductivity by the electric field.

To compare our measurements with other possible measurements of the supercurrent noise, we converted the magnitude time traces to the traces of the frequency fluctuations $f(t)$ (green traces in Fig. 2a), by dividing the former by a gradient of the resonance curve at the working point. The frequency fluctuations $f(t)$ were converted into the noise spectra by doing Fourier transform. Figure 2b shows examples of the frequency noise spectra for four distinct gate voltages. At low $V_G$, the noise spectrum is almost frequency independent, i.e., white noise. However, as $V_G$ increases the noise changes both quantitatively and qualitatively, and, importantly, it grows faster at low frequencies such that the slope of the frequency dependence increases with $V_G$. The noise of the resonance frequency is associated with the noise of the Josephson inductance of the Dayem bridge, which is proportional to the critical current noise[23,24]. Under the assumption that the application of $V_G$ does not change thermal equilibrium of the nanoconstriction[4–7], the increment of the supercurrent noise power with $V_G$ contradicts the theoretical prediction that the integrated supercurrent noise should decrease as $I_c^2$ [23]. Such an observation leads us to the conclusion that the gated nanobridge is either locally overheated with respect to the environment or driven out of equilibrium by the injected high-energy quasiparticles[9–11,25].

In order to quantify the noise spectra at different gate voltages, we fitted the functional form, $Af^{-k}$, where $k$ represents the slope of the spectra, to the measured data (see Fig. 2b). The resulting dependence of the obtained exponent $k$ as a function of $V_G$ is shown in the inset of Fig. 2b. At low gate voltages $|V_G| < 25$ V, the value of $k$ is close to zero, but it rapidly increases and approaches $k \approx 1$ at higher voltages. If the gated nanobridge remains in thermal equilibrium, then, according to the fluctuation-dissipation theorem (FDT)[26], the noise increase can be explained by either the rise of the nanobridge temperature, or increase of dissipation. Even if the system is driven out of thermal equilibrium, a

generalised version of the FDT holds[27], so does the conclusion about the higher temperature or greater dissipation in the system. Thus, our noise measurements reaffirm that the suppression of the critical current is not an electrostatic field effect[4–7]. It is rather a nonequilibrium state of a superconductor, which is caused by high-energy quasiparticle injection[9–11,25].

Whether or not the system is in the equilibrium state can be revealed via histograms of the measured noise of $I$ and $Q$ components. Figure 3a shows the two-dimensional probability density function (PDF) at $V_G = 20$ V and $V_G = 50$ V and its projections on the $I$ and $Q$ axes. The clear asymmetry of the PDF at high voltage is in strong contrast with the symmetric PDF at low voltage. The latter, as expected, looks like an ideal circle on the two-dimensional plane and the two projections are perfect Gaussian functions. This proves that the system is in thermodynamic equilibrium[26]. The distorted histograms at high gate voltage are evidence that the equilibrium is broken. To determine the point at which the system is no longer in thermodynamic equilibrium, we quantify the non-Gaussianity using the 3rd moment (skewness, $\mu_3$) of the distribution normalised to the standard deviation, $\sigma_f$, namely, $\mu_3/\sigma_f^3$. A non-zero value of the skewness shows that the distribution is not Gaussian, and the system is out of thermal equilibrium. In addition to skewness, we present an analogue of the signal-to-noise ratio, viz., normalised frequency shift $\Delta f/\sigma_f$. The gate dependence of both quantities is presented in Fig. 3b. Both panels show that at low gate voltages, $|V_G| < 25$ V, the system stays in thermodynamic equilibrium, but outside this range, the normalised frequency shift increases and skewness becomes negative. This supports our conclusion about the nonequilibrium state of the nanobridge under the applied high gate voltage.

The nonequilibrium distribution of quasiparticles should be less pronounced at higher temperatures owing to the stronger electron–phonon coupling[28] and our experiments at different temperatures confirm this. Figure 4 summarises the gate voltage dependence of all obtained quantities at different temperatures. All the six panels have a common feature, a plateau, in the range of ~±25 V at 10 mK with no dependence on $V_G$. This plateau becomes wider at higher temperatures. It is worth noting that the dependence of $\Delta f_0$ and $Q_i$ on $V_G$ shown in Fig. 4a, b, respectively, bear resemblance to the dependence of the critical current reported in earlier works (see, for example, Fig. 2b in ref. [4] and Fig. 2b in ref. [5]). A similar behaviour, where the plateau region widens with temperature, is also observed for the exponent $k$, the

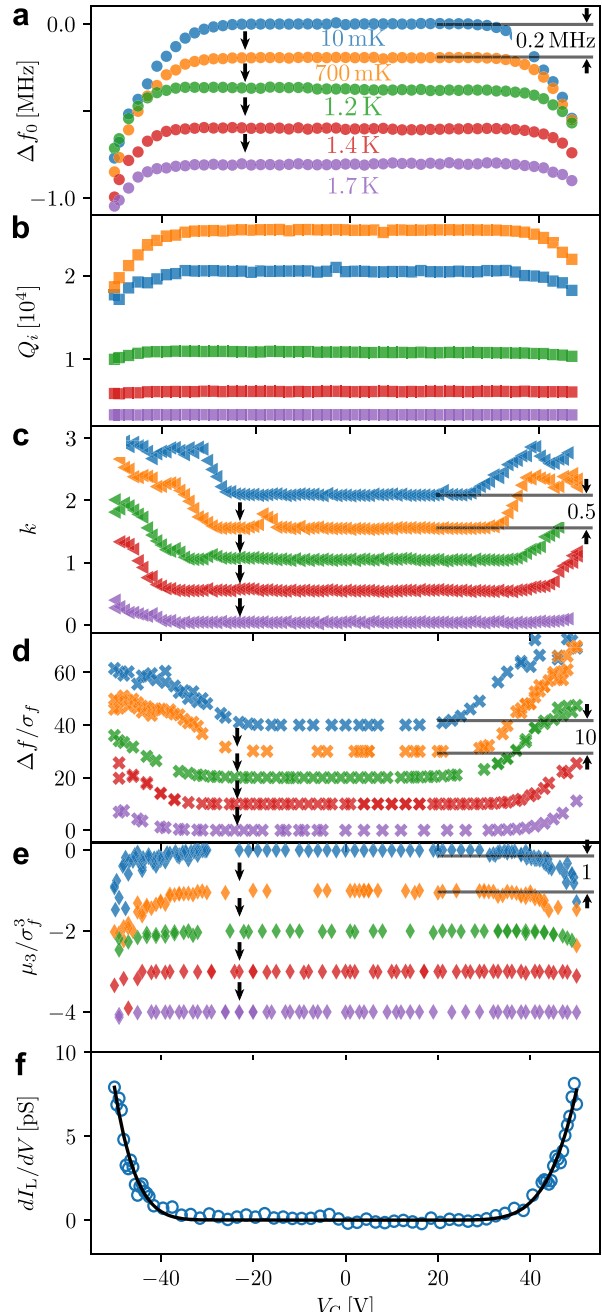

**Fig. 4 Summary of the gate dependencies for all measured quantities at different temperatures.** Colours represent the cryostat temperature, which is common across all panels. For clarity, curves have been equidistantly shifted from each other, shown by the black arrows, by the value shown in black on the right of the corresponding panel. **a** The shift of the resonance frequency; **b** internal quality factor; **c** exponent $k$ of the noise PSD $\propto f^{-k}$; **d** normalised frequency shift, and **e** normalised 3rd moment (skewness) of the frequency fluctuations. **f** Measured differential conductance $dI_L/dV_G$ of the gap between the gate and Dayem bridge overlapped with a fit obtained using Eq. (3).

normalised frequency shift $\Delta f/\sigma_f$ and the normalised skewness $\mu_3/\sigma_f^3$, shown in Fig. 4c–e, respectively. Such dependence on temperature suggests that the observed effect originates from gate voltage-controlled dissipation in the constriction as higher gate voltage is required in order to see a change in the measured

quantities at higher temperatures. All the data presented here point to the existence of an additional source of dissipation that results in higher noise and a distorted distribution function at higher $V_G$. The origin of such behaviour can be attributed to leakage between the gate and constriction. Figure 4f presents the measured dependence of the leakage differential conductance on the gate voltage. Similarly to the dependence of the frequency shift, quality factor, noise, etc., it has a plateau below about $|V_G| \sim$ 30 V that is followed by a stronger dependence above this value. This indicates that the gate leakage and all other observations are correlated.

## Discussion

It is well known that electrons, when they are given enough energy, can escape from a metal surface in a process known as electron emission[29]. One possible mechanism is escape of an electron under an intense electric field, that is, the field emission, where the energy supplied by the field is greater than the barrier height or work-function, which have typical values of a few eV. The gate voltages applied in our experiments and in refs. [4–7] reach tens of volts on the length scale of ~100 nm. Such high voltages correspond to an electric field strength of hundreds of MV m$^{-1}$, which is comparable with the dielectric strength of ~1 GV m$^{-1}$ in nanoscale vacuum gaps[30] and exceed the dielectric strength of SiO$_2$ and sapphire. Despite the fact that the leakage of a similar magnitude was reported in previous publications[4–7], their authors alleged such fields insufficient to cause the emission. We note that the superconducting state of electrodes does not play any role as the characteristic super-conducting energy gaps are of the order of 1 meV or below. Furthermore, the violation of bipolarity of the injected current is not expected to be observable since the current flowing through the constriction exceeds the leakage current by many orders of magnitude. The noise properties may be showing a slight asymmetry with respect to the applied gate voltage but are beyond the scope of this manuscript.

The electron emission current $I_L$ as a function of the applied voltage $V_G$ can be approximated within the Fowler–Nordheim model as[31]

$$I_L = \alpha V_G^2 \exp\left(-\frac{\beta}{V_G}\right),\tag{3}$$

where $\alpha$ and $\beta$ are some coefficients that depend on material properties and geometry. We differentiated Eq. (3) and fitted the resulting expression to the measured differential conductance of the gap between the gate and the nanobridge, $dI_L/dV_G$. Figure 4f shows that the agreement between the experimental data and fit is excellent. The fitting parameters are: $\alpha = 1.5 \pm 0.4$ nAV$^{-2}$ ($2.2 \pm 1.2$ nAV$^{-2}$) and $\beta = 0.39 \pm 0.01$ kV ($0.54 \pm 0.03$ kV) obtained for positive (negative) bias. The values of $\beta$ give us the barrier height of 0.68 eV and 0.85 eV for the positive and negative bias, respectively. As expected, the barrier height is smaller than the workfunction of vanadium 4.3 eV[32]. The asymmetry of the tunnel barrier resulting in the different height values for the positive and negative bias can presumably be explained by the asymmetry of the device. The exact value of the barrier height may depend on many factors and its discussion is beyond the scope of this paper. We have used Eq. (3) to fit the current–voltage characteristics presented in refs. [4,5] and also found an excellent agreement.

The experimental data suggest that a miniscule leakage current between the gate and constriction generated through the field emission mechanism and described by the Fowler–Nordheim model[31] is responsible for the observed out-of-equilibrium effects. Indeed, each electron arriving at the nanobridge carries energy of tens of eV, which is sufficient to destroy tens of thousands of

Cooper pairs and generate quasiparticles locally. The average dissipated power at the nanobridge produced by the leakage current of ~10 pA at tens of volts can exceed 100 pW and is sufficient to cause observed out-of-equilibrium effects or raise the effective temperature above 1 K. Therefore, based on the presented experimental observations and a strong correlation between the gate voltage dependence of the leakage current and measured quantities such as the shift of the resonance frequency, the quality factor and the noise properties of the system, we conclude that the suppression of the critical current in Dayem bridges reported in this manuscript and the earlier works[4–7] is caused by the injection of high-energy electrons at intense electric fields resulting in an out-of-equilibrium thermodynamic state of the constriction. Although the electron emission is thought to be ruled out in ref. [33], the interpretation involving the quasiparticle injection, rather than the electrostatic field effect, is also supported by the recent experimental works[18,19] and agrees with the existing theories.

## Methods

**Device fabrication**. First, a 30 nm thick vanadium film was deposited at a rate of 5 Å s$^{-1}$ in an electron-gun evaporator on a Si wafer covered with a 300 nm thick thermally grown oxide. The base pressure and deposition pressure in the vacuum chamber were in the range of $1 \times 10^{-7}$ mbar to $2 \times 10^{-7}$ mbar. Then, the wafer was spin-coated with a positive tone electron resist, which was exposed in an electron-beam writer JEOL JBX-5500 at an acceleration voltage of 50 kV. After the resist development, the required pattern was formed in the resist layer, which was used as an etching mask in the Oxford Instruments PlasmaPro 100 inductively coupled plasma etcher. Device fabrication was completed by etching vanadium in Cl$_2$ at a flowrate of 20 sccm, pressure of 5 mbar, and microwave power of 200 W.

Imaging of the fabricated chips was performed in a field-emission scanning electron microscope JEOL JSM-7800F at an acceleration voltage of 10 kV.

**Device characterisation at low temperatures**. Device characterisation was carried out in a cryogen-free dilution refrigerator BlueFors BF-LD250 capable of reaching temperatures below 10 mK. The refrigerator was equipped with DC and microwave coaxial line lines. Resonator response curves and noise time traces were measured using a vector network analyzer (Agilent E5071C). Noise spectra were measured with a signal analyzer (Agilent N9030A).

## Data availability

Data files and codes are available at https://doi.org/10.17635/lancaster/researchdata/436.

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

## Acknowledgements

We thank D.V. Averin, A. Braggio, F. Giazotto, I. Khaymovich, O. Kolosov, A.S. Melnikov, A.D. Zaikin and all members of the Lancaster University ULT group for fruitful discussions. This research was supported by the UK EPSRC grants no. EP/P022197/1, EP/P024203/1, the Royal Society International Exchanges scheme (grant IES\R3\170054) and the EU H2020 European Microkelvin Platform (grant agreement 824109). This work was partially funded by the Joint Research Project 17FUN10 ParaWave of the EMPIR Programme co-financed by the Participating States and from the European Union's Horizon 2020 research and innovation programme.

## Author contributions

The experiment was conceived by S.K. and Yu A.P. The samples were fabricated by S.K., A.G. and I.G. Low-temperature measurements were carried out by A.G., I.G. and S.K. Data analysis was done by A.G., I.G., S.K. and V.T. All authors contributed to the interpretation of the results. The manuscript was mainly written by A.G., S.K., Yu A.P. and V.T.

## Competing interests

The authors declare no competing interests.
