## [Peer Review File · Nature Communications]

REVIEWER COMMENTS

Reviewer #1 (Remarks to the Author):

I found this paper very interesting and pleasant to read, this is a very interesting Communication. The authors fabricated electrically gated Dayem bridges inside a cleverly designed resonating structure, which allowed them to apply large gate voltages to the bridge without leaking microwaves out. The authors found a clear evidence that applying voltage of order a few 10's of Volts heats up the bridges via electron tunneling. This result disproves the idea from the previous work by Giazotto's group that there might be a non-dissipative superconducting field effect, which indeed seemed too good to be true. The authors also investigated frequency noise in the out-of-equilibrium state, which adds value to their evidence for heating via electron tunneling.

Compared to previous transport/tunneling experiments on this topic, this radio-frequency experiment is a lot more valuable because it directly demonstrates that making quantum devices based on heating of Dayem bridges is probably a bad idea. Not surprisingly, the quality factor rapidly drops. I wish that more papers focus on a specific scientific target (here to test Giazotto's group controversial hypothesis) and avoid blowing out of proportions the potential applications of their devices.

I wonder if the Authors could comment on why the possibility of electronic emission was rejected in the earlier work and could it be that some junk on the chip surface enhances the leakage current.

In conclusion, I recommend the paper for publication at Nature Communications.

Reviewer #2 (Remarks to the Author):

This manuscript addresses the so-called electrostatic field effect in superconductors. This is an issue which has received considerable attention since the observations from the Pisa group (Refs. [4-7] in the manuscript) suggesting the possibility to control the critical current of a superconducting nanowire using electrostatic gates.

In contrast to the statements from the Pisa group on the origin of this effect, recent works from groups in Zurich and Harvard (Refs. [29] and [30] in the manuscript) point out to the injection of high energy quasiparticles through leakage currents from the gate electrodes to the superconductor as the source of the observed effect. The present manuscript provides more evidence to this explanation using alternative methods.

In particular, the authors analyze the change in the wire superconducting properties by their effect on an inductively coupled superconducting resonator. Both the decrease of the resonator frequency and its quality factor as the applied gate voltage increases exhibit a clear correlation with the measured leakage currents. Additionally, the measured noise properties indicate that a non-equilibrium distribution of quasiparticles is established, consistent with an overheating of the nanowire.

In my opinion the results in this manuscript are sound and deserve to be published. I have, however, a couple of comments that the authors and the editors should consider:

- In my opinion the "field-effect" issue has become an artificial "hot-topic", mainly due to the reluctance of the Pisa group to accept the evidence on overheating provided by Refs. [29] and [30], which is further supported by the present work. This is demonstrated by their most recent publication (Ref. [28] in the manuscript) where, even with similar data on leakage currents, they discard the quasiparticle injection as the origin of the effect.

- I believe that the authors should cite Refs. [29] and [30] at the beginning of their manuscript as these have been the first pointing out the high-energy quasiparticle injection mechanism which is further supported in this work.

Reviewer #3 (Remarks to the Author):

The authors present an excellent experimental work to elucidate the origin of the critical current decrease observed in gated metallic Josephson weak-links but not fully understood yet. To do so they measure the voltage induced changes in the resonance frequency and quality factor of a superconducting LC-resonator ended by a gated Dayem bridge. These changes are related to variations of the real (resistance) and imaginary (inductance) parts of the Josephson impedance. This method allows one to measure the Josephson impedance at high frequency (here 3.3 GHz) with unsurpassed resolution. For gate voltage larger than 30 V, the quality factor and the resonance frequency decrease. The authors observe that this decrease is associated with an increase of both the signal noise and the leak current from the gate, they conclude that the reduction of the critical current is due to out-of-equilibrium superconductivity and not to electrostatic field effect.

While this conclusion is supported by the experiment, the manuscript contains a series of claims that are incorrect and should be removed before publication. In particular the authors wrongly confuse overheating and out-of-equilibrium effects (an out-of-equilibrium state is not necessarily defined by an effective temperature). While their work supports out-of-equilibrium superconductivity induced by a current leak from the gate, it doesn't provide any evidence that such a leak produces an effective temperature. Instead, all the experimental results show that the data can not be explained by a simple increase in the quasiparticle temperature. To be more precise : 1) The amplitude of the gate induced noise goes as $1/f$ at low frequency (see Fig2) while thermal noise is white ; 2) The I-Q resolved noise (see Fig.3) is not gaussian while thermal noise is expected to be gaussian as also shown for the data measured at $V_g=20$ V where the leak current is negligible; 3) At higher bath temperatures (see Fig.4) but zero gate voltage not extra noise is measured in the transmission coefficient (either in the amplitude or in the I/Q signals). Therefore the data reported in the manuscript strongly support a more interesting out-of-equilibrium state induced by the leak current rather than a thermal one. The manuscript should be revised accordingly.

The authors claim that the leak current is actually due to field-emission and it's well described by the Fowler-Nordheim model (eq. 3). It would be interesting to know if the value of the α and β parameters obtained by the fit are consistent with geometry of the device.

Finally as a minor remark, I think that figure 3a is a bit misleading as the difference on the width of the distributions at $V_g=20$ V and $V_g=50$ V are not evident. I suggest to remove the normalization by the standard deviation of the PDF and to add on red, the same data measured at $V_g=20$ V (this should give a very little circle in the middle of the blue area).

Comments to reviewer 1

1. *I wonder if the Authors could comment on why the possibility of electronic emission was rejected in the earlier work and could it be that some junk on the chip surface enhances the leakage current.*

This is a very interesting question. The authors of the original publication Nature Nanotechnology **13**, 802 (2018) did observe the rapid increase of the leakage current at high gate voltages (see. Supplementary information in the arXiv version of the paper <https://arxiv.org/pdf/1710.02400.pdf>). Moreover, they have also presented the leakage current vs. gate voltage dependence in Nano Letters **18**, 4195 (2018), but they completely ruled out quasiparticle injection through the electron field emission mechanism.

Here is a quote from their Nature Nanotechnology (2018) paper: “Moreover, any hot spot mechanism due to direct charge injection into the wire can be ruled out as the main driving principle for I_c suppression due to the incompatibility with the bipolarity of the effect, and the independence of the critical temperature on gate voltage. In addition, a possible electron field-emission mechanism can also be excluded, because it is usually expected to occur for electric fields much larger than those applied in the experiment.” The first argument implies that, depending on the polarity of the applied gate voltage, the injected current is either added to or subtracted from the current flowing through the constriction, thus resulting in the asymmetric effect. This is true, however, the added or subtracted current is many orders of magnitude smaller than the constriction current (pA vs μ A), so one cannot expect this effect to be visible. The second argument assumes that the applied electric field is not strong enough to cause a breakdown. The handbook value of the dielectric strength for a typical insulating material is in the range 1 MV m^{-1} to 10 MV m^{-1} . Taking a gap between the gate and constriction of 100 nm and using a characteristic gate voltage of 10 V, we obtain the field strength of 100 MV m^{-1} , well above the dielectric strength of SiO_2 or sapphire (up to 40 MV m^{-1}). However, the key parameter here is not the leakage current itself, which is negligible, but the enormous energy of the injected quasiparticles resulting in more quasiparticle excitations and suppression of superconductivity in the constriction. The common feature of all observations reported by the Pisa and other groups and also confirmed in our experiments, is that the onset of suppression of the critical current correlates with the rapid rise of the leakage current.

Just comparing the data reported by several groups, one can see that the critical gate voltage at which the supercurrent is completely suppressed varies a lot. When a nanowire is fabricated on a bare Si substrate (<https://arxiv.org/abs/2005.00462>), the critical voltage can be as low as 2.8 V corresponding to the electric field of 35 MV m^{-1} assuming a gate-nanowire gap of 80 nm. However, when a sapphire or SiO_2 substrate is used, even 70 nm is not enough to suppress the critical current completely (Nature Nanotechnology **13**, 802 (2018); ACS Nano **13**, 7871 (2019)), which gives a critical field strength in excess of 700 MV m^{-1} for a gap of 100 nm. On the other hand, for two identical samples (I_c 's are $28 \mu\text{A}$ and $24 \mu\text{A}$) fabricated in the same run on SiO_2 substrate (Nano Letters **18**, 4195 (2018)), the critical voltages were very different: 32 V and 8 V, respectively. A factor of four variation of the critical voltage may be attributed to the junk on the chip surface near one of the nanowires, so the junk hypothesis might be valid. We should add though that the overall statistics on the critical voltage is rather small and it is too early to make conclusions about the reasons for its variations.

Comments to reviewer 2

1. *In my opinion the “field-effect” issue has become an artificial “hot-topic”, mainly due to the reluctance of the Pisa group to accept the evidence on overheating provided by Refs. [29] and [30], which is further supported by the present work. This is demonstrated by their most recent publication (Ref. [28] in the manuscript) where, even with similar data on leakage currents, they discard the quasiparticle injection as the origin of the effect.*

We agree that the Pisa group has been dismissive of the evidence on local overheating and presence of dissipation. There is a clear correlation between the growth of the leakage current and suppression of the critical current. Interestingly, even if a superconducting nanowire is suspended, the critical voltage remains comparable to that observed for non-suspended devices. This may be due to the fact, that the leakage current flows not through the vacuum gap directly to the suspended part of the device, but to a non-suspended part through the substrate. We argue that injection of high-energy quasiparticles through electron field emission is the most probable mechanism of the suppression of the critical current.

In their most recent publication (M. Rocci, *et. al*, ACS Nano, **14**, 12621, (2020)), the Pisa group mistakenly assumed that “... cold-electron emission generally requires E at least of the order of 1 GV m^{-1} to 10 GV m^{-1} ...”. This led them to ruling out the cold electron emission as the origin of the observed leakage currents, as the latter appeared at an order of magnitude smaller fields.

2. *I believe that the authors should cite Refs. [29] and [30] at the beginning of their manuscript as these have been the first pointing out the high-energy quasiparticle injection mechanism which is further supported in this work.*

We have cited Refs. [29] and [30] on page 1. Now they are cited as [9] and [10].

Comments to reviewer 3

1. *While this conclusion is supported by the experiment, the manuscript contains a series of claims that are incorrect and should be removed before publication. In particular the authors wrongly confuse overheating and out-of-equilibrium effects (an out-of-equilibrium state is not necessarily defined by an effective temperature). While their work supports out-of-equilibrium superconductivity induced by a current leak from the gate, it doesn't provide any evidence that such a leak produces an effective temperature. Instead, all the experimental results show that the data can not be explained by a simple increase in the quasiparticle temperature. To be more precise :*
 1. *The amplitude of the gate induced noise goes as $1/f$ at low frequency (see Fig 2) while thermal noise is white;*
 2. *The I-Q resolved noise (see Fig.3) is not gaussian while thermal noise is expected to be gaussian as also shown for the data measured at $V_g = 20$ V where the leak current is negligible;*
 3. *At higher bath temperatures (see Fig.4) but zero gate voltage not extra noise is measured in the transmission coefficient (either in the amplitude or in the I/Q signals). Therefore the data reported in the manuscript strongly support a more interesting out-of-equilibrium state induced by the leak current rather than a thermal one. The manuscript should be revised accordingly.*

We agree that a system driven out of equilibrium cannot be characterised by an effective temperature. We have corrected the manuscript in all corresponding places.

2. *The authors claim that the leak current is actually due to field-emission and it's well described by the Fowler-Nordheim model (Eq. 3). It would be interesting to know if the value of the alpha and beta parameters obtained by the fit are consistent with geometry of the device.*

We have added the values of the two fitting parameters and corresponding discussion below Eq.(3). Both alpha and beta can be used to extract the effective barrier height, however, one needs to know the cross-sectional area of the current path in order to obtain the barrier height from alpha. Hence, we used only beta for this purpose. Assuming the gap between the constriction and the gate to be about 100 nm (which might be an overestimate), we obtain the barrier height of about 0.68 eV and 0.85 eV for positive and negative bias, respectively. As expected, these values are lower than the Ti work function, 4.3 eV. More detailed investigation into the barrier properties is beyond the scope of this work.

3. *Finally as a minor remark, I think that figure 3a is a bit misleading as the difference on the width of the distributions at $V_g = 20$ V and $V_g = 50$ V are not evident. I suggest to remove the normalization by the standard deviation of the PDF and to add on red, the same data measured at $V_g = 20$ V (this should give a very little cercle in the middle of the blue area).*

We have altered the figure as suggested.